# SIMPLICITY BIAS IN OVERPARAMETERIZED MACHINE LEARNING

## ABSTRACT

A thorough theoretical understanding of the surprising generalization ability of deep networks (and other overparameterized models) is still lacking. Here we demonstrate that *simplicity bias* is a major phenomenon to be reckoned with in overparameterized machine learning. In addition to explaining the outcome of simplicity bias, we also study its *source*: following concrete rigorous examples, we argue that (i) simplicity bias can explain generalization in overparameterized learning models such as neural networks; (ii) simplicity bias and excellent generalization are optimizer-independent, as our examples show, and although the optimizer affects training, it is not the driving force behind simplicity bias; (iii) simplicity bias in pre-training models, and subsequent posteriors, is universal and stems from the subtle fact that *uniformly-at-random constructed* priors are not *uniformly-at-random sampled*; and (iv) in neural network models, the biasing mechanism in wide (and shallow) networks is different from the biasing mechanism in deep (and narrow) networks.

## 1 INTRODUCTION

Contemporary practice in deep learning has challenged conventional approaches to machine learning. Specifically, deep neural networks are highly overparameterized models with respect to the number of data points and are often trained without explicit regularization. Yet they achieve state-of-the-art generalization performance. It is interesting to note that these observations are not limited just to neural network models. Qualitatively similar behaviour has also been observed when using boosting with decision trees and random forests (Wyner et al., 2017) and other non-network learning problems, with some examples dating back to the early 1990s (Loog et al., 2020).

A thorough theoretical understanding of the unreasonable effectiveness of deep networks (and other overparameterized models) is still lacking. Previous work (e.g., (Neyshabur et al., 2014; 2018)) has suggested that an implicit regularization is occurring in neural networks via an implicit *norm minimization*; in particular, the minimization of the (generalized) norm was conjectured to be a by-product of the "optimizer", the method by which the network is trained (i.e., stochastic gradient descent, SGD). However, this has been questioned by theoretical and empirical work showing strong evidence to the contrary (Razin & Cohen, 2020; Huang et al., 2019; Kawaguchi et al., 2017). Furthermore, although the reasoning behind SGD as an implicit regularizer (Hochreiter & Schmidhuber, 1997; Keskar et al., 2016) is insightful ("solutions that do not generalize well correspond to *sharp minima*, and added noise prevents convergence to such solutions"), there are examples where a good generalization is obtained irrespective of the optimizer used (Huang et al., 2019; Kawaguchi et al., 2017; Wu et al., 2017).

Here we propose an entirely different and new approach: instead of implicitly assuming that learning-models are uniformly-probable random objects (prior to training), we suggest that **the probability space over models is in fact biased towards simple functions**. Thus, the trained model will likely extrapolate well without fluctuating wildly in-between the training data-points.

Despite some preliminary results (Arpit et al., 2017; Valle-Perez et al., 2018; Mingard et al., 2019; De Palma et al., 2019), a theoretical treatment with a rigorous proof of such a bias is still lacking. Here, in addition to explaining the *outcome* of simplicity bias, we also study its **source**. This work suggests that, for typical overparameterized models (in addition to explaining generalization),

simplicity bias is in fact universal and nearly unavoidable. The crux of the matter is that the functions obtained (at initialization) are *uniformly-at-random constructed* but not *uniformly-at-random sampled*. In other words, the naive "default" probability space over functions, where each unique function is sampled with equal probability, is actually conceptually wrong and irrelevant for this domain. Instead, it is well-known that if a *construction process* is used to "sample" complex objects then the resulting probability space is *not* a uniform space (even if the construction itself is uniformly random, i.e., in each sub-step we choose uniformly from several construction options). Moreover, different (spaces of) construction processes yield different probability spaces over functions. However, several properties can be expected to be universal in some sense, with simplicity bias being a case in point (Chauvin et al., 2004).

Below we present a concrete rigorous example; we then identify two distinct mechanisms that bias towards simplicity and provide further theoretical evidence to dispute the relevance of focusing on shallow (and wide) networks while researching deep (and narrow) networks.

## 2 RESULTS

We now describe three concrete examples: overparameterized learning of a Boolean function, infinitely-wide neural networks (where we revisit known results), and infinitely-deep **non-wide** neural networks.

**Learning a Boolean function**. Let $\mathcal{F}_n$ be the set of all Boolean functions on $n$ variables. There are $2^n$ possible binary inputs, and thus there are $2^{2^n}$ such Boolean functions, $|\mathcal{F}_n| = 2^{2^n}$.

For $f \in \mathcal{F}_n$, let $(X, Y_f)$ denote a uniformly random sample of observations regarding $f$, and $|(X, Y_f)|$ denote the sample size (i.e., the number of input–output pairs observed).

How likely are we to overfit when fitting $\hat{f}$ based on a small sample? Consider the following negative claim:

*Claim 1. The vast majority of functions that agree with $f$ over the sample obtain only chance agreement with $f$ out-of-sample (i.e., agreement $\approx 1/2$ out-of-sample).*

More precisely, for a fixed sample size and a fixed $\epsilon > 0$, the law of large numbers immediately entails that if we choose a function $\hat{f}$ completely at random (subject to fitting the sample) then the probability of agreement on more than $1/2 + \epsilon$ out-of-sample goes to 0 as $n \to \infty$ (note that $f$ need not be fixed).

However, our main claim here is that this negative result is actually misleading and demonstrates a misguided way of thinking. It is true that when fitting by *choosing a function* at random we are likely to overfit; however, in practice we seldom choose a function at random, instead we *construct a function* at random or choose a *representation* at random. In particular, Boolean functions are typically implemented using circuits or binary AND/OR trees. Thus, a more nuanced question would be: if we choose a Boolean circuit $t$ completely at random (subject to fitting the sample) what is the probability of agreement on more than $1/2 + \epsilon$ out-of-sample?

Let $\mathcal{T}_{m,n}$ denote the set of all binary Boolean trees with $m$ internal nodes over $n$ input variables (and consider large and "overparameterized" trees with $m \gg n$). Formally, a uniform probability space over $\mathcal{T}_{m,n}$ was introduced in (Chauvin et al., 2004) in the following manner: choose uniformly at random a rooted binary tree and label its $m$ internal nodes randomly with AND and OR, and the $m + 1$ external nodes with a literal, i.e., a variable or its negation. Each of the $m$ inner nodes is labeled with AND or OR with equal probability $1/2$ and independently of the other nodes; each leaf is labeled with a literal, chosen according to the uniform distribution on the $2n$ literals and independently of the labeling of all other nodes.

Again, if we construct a tree $t_i \in \mathcal{T}_{m,n}$, we can find many candidates that agree with $f$ perfectly over the sample $(X, Y_f)$ while agreeing with $f$ over only $\approx 1/2$ of the out-of-sample data (i.e., chance agreement). Therefore, you might surmise that the following learning algorithm is (statistically) ill-advised:

```
Naive Algorithm.  Given (X, Y_f):

    1. Construct uniformly at random a tree t_i ∈ 𝒯_{m,n}.
    2. Check if f_i, the function corresponding to t_i, agrees with f
       over (X, Y_f).  If so, return f_i.  If not, go to step 1.
```

Nevertheless, we have the following surprising result linking the complexity[1] of $f$, the sample size, and $\epsilon$ (generalization):

**Theorem 1 (Random trees interpolate without overfitting)** *Let $L_f$ denote the complexity of $f$, and $s$ the sample size, $s = |(X, Y_f)|$. For a given $0 < \epsilon < 1/2$, fix $b$ such that $\log(1/2 + \epsilon) < -b$. As $n \to \infty$, if $L_f \leqslant b\frac{s}{\log n}$ then with high probability the output of the naive algorithm:*
*(i) agrees with $f$ completely over the sample, and*
*(ii) agrees with $f$ over more than $1/2 + \epsilon$ of the out-of-sample data.*

*Remark*: note that $\epsilon$ and $b$ are fixed, but $L_f$ and $s$ (and $m$) are not.

*Proof.* According to theorem 3.1 of Lefmann & Savický (1997)[2], the probability of randomly constructing a tree that corresponds to $f$ is $\geqslant \frac{1}{4}\left(\frac{1}{8n}\right)^{L_f}$. Therefore, the number of attempts by the naive algorithm is dominated by a geometric random variable with mean $4(8n)^{L_f}$, and a perfect fit to the data is found in $O_p\left((8n)^{L_f}\right)$ attempts.

The functions in $\mathcal{F}_n$ can be partitioned according to their agreement with $f$:

- Poor or chance agreement. Each function in this class has a probability of less than $1/2 + \epsilon$ of agreeing with $f$ for a random input.
- Adequate agreement (or better) [3]. Each function in this class has a probability of more than $1/2 + \epsilon$ of agreeing with $f$ for a random input.

The naive algorithm results in overfitting if it samples a function $\hat{f}$ from the first class (the poor/chance agreement class) which agrees with $f$ over the sample data; however, for a given $\hat{f}$, since the sample data is sampled uniformly at random, the probability of such an agreement is $\leqslant (1/2 + \epsilon)^s$. Finally, since the naive algorithm performs $O_p\left((8n)^{L_f}\right)$ attempts, an application of the union bound provides the following upper bound on the probability of overfitting:

$$P(overfit) = O_p\left((1/2 + \epsilon)^s (8n)^{L_f}\right) \tag{1}$$

but now

$$(1/2 + \epsilon)^s (8n)^{L_f} \leqslant (1/2 + \epsilon)^s (8n)^{b\frac{s}{\log n}} =$$

$$= \left((1/2 + \epsilon)(8n)^{\frac{b}{\log n}}\right)^s$$

and bearing in mind that $\lim n^{\frac{b}{\log n}} = e^b$, we get that the right-hand side is $o(1)$ for large $s$. ⊠

The following example illustrates the point:
*Example 1.* Consider a binary classification task of black/white images with $n = 28 \times 28 = 784$ pixels (with $2^{2^{784}} \approx 10^{10^{235}}$ classifiers to choose from). If we provide the naive algorithm a sample of $s = 10^6$ images, when will it avoid overfitting? Theorem 1 says that overfitting is avoided if $L_f = O(s/\log n)$, in other words: if $f$ can be implemented using an order of

---

[1]Here, following (Chauvin et al., 2004) we define the complexity of a Boolean function, $f$, as $L_f = $ minimal size of a tree computing $f$, where the *size* of a tree is the number of internal nodes it has.

[2]Note we use the notations of Chauvin et al. (2004), not Lefmann & Savický (1997). See also theorem 1 in Chauvin et al. (2004).

[3]It is possible to further add here an *almost sure agreement* class, agreeing with $f$ with probability $1 - o(1)$. A similar proof, with additional bookkeeping beyond the scope of this note, can show that for a large enough sample size (or simple enough $f$) the naive algorithm will provide $\hat{f}$ almost surely agreeing with $f$.

$s/\log n = 10^6/\log 784 \approx 0.35 \times 10^6$ gates, which seems quite a lot and is very permissive.

**Wide neural networks**. In a very wide network (see Fig. 1A) with random weights, the situation is in fact straightforward: under the appropriate conditions, the output layer simply sums up a large number of random variables and thus the central limit theorem kicks in. The result is that the network is a Gaussian process (at initialization), which is a simple and well-behaved function of the input. Moreover, previous work regarding very wide networks (Jacot et al., 2018) demonstrated this Gaussian process and further found that after training via gradient descent the result is akin to Gaussian process regression (Jacot et al., 2018) (i.e., a Gaussian process conditioned on interpolating the sample data; see (Williams & Rasmussen, 2006)). Denote by $\mathcal{B}_{train}(\cdot)$ a Gaussian process (whose mean function and covariance are training-data dependent) conditioned on passing through the sample points[4]. From ref. (Jacot et al., 2018) we can summarize the following proposition:

**Proposition 1** *In the setting of ref. (Jacot et al., 2018), a wide network trained on a noise-free data* $(X, Y_f)$ *via gradient descent after a random initialization is distributed as* $\mathcal{B}_{train}(\cdot)$.

*Remark*: Needless to say, the mean function and covariance are training-data dependent, but given $(X, Y_f)$ they are deterministic. The only randomness is due to the random Gaussian initialization of the network weights.

Consider again the following naive "training" algorithm:

```
Naive Algorithm (for neural networks).  Given (X, Y_f):

   1. Initialize the weights of a network (f̂) at random from a
      normal distribution.
   2. Check if the network f̂ agrees with f over (X, Y_f).  If so,
      return f̂.  If not, go to step 1.
```

*Remark*: This algorithm is obviously non-constructive. In particular, the probability of agreement is zero. However, (i) it could be modified to check if the disagreement is smaller than a predefined threshold (ii) efficiency is not the issue here, rather the following question: what could be said about the result if the algorithm does stop?

**Proposition 2** *In the setting of ref. (Jacot et al., 2018), a wide network trained on a noise-free data* $(X, Y_f)$ *via the naive algorithm (given the algorithm has stopped) is distributed as* $\mathcal{B}_{train}(\cdot)$.

*Remark*: Here too the randomness is due to the random Gaussian initialization of the weights.

This highlights that it is not gradient-based training which contribute to the statistical efficiency and generalization of the outcome (although GD is indeed important for computational efficiency - and for speeding up training). Rather, there is an inherent simplicity bias due to the random construction of the network via random weights initialization.

However, the main driving force for the emergence of these simple functions in the setting above is the large *width* of the network, and not its depth (Lee et al., 2019). For a "narrow" but *deep* neural network we show below an entirely different mechanism which produces simplicity bias nevertheless.

**Deep neural networks**. In a multi-layered network (see Fig. 1B) each layer can be viewed as an operator in a dynamical system that acts on the output of its preceding layer. Under the appropriate conditions, this should lead to convergence to a fixed point[5] regardless of the initial input - i.e. a simple "constant function".

*Example 2*. Consider a standard fully connected ReLu network with no bias, and Gaussian

---

[4]The notation $\mathcal{B}$ is due to the analogy to the Brownian bridge process.

[5]The notion of a fixed point is more subtle in the case of *random* dynamical systems, but similar behavior is expected nonetheless (Boxler, 1995; Bhattacharya & Majumdar, 2007).

weights initialization; i.e., prior to training, the output of $x_{j,l}$, the $j^{th}$ unit in the $l^{th}$ layer, is $R\left(\sum_{k=1}^{w} a_{j,k}^{(l)} x_{k,l-1}\right)$, where $w$ is the width of the network, each weight $a_{j,k}^{(l)}$ is a standard normal random variable independent of the rest, and $R(x) = \max(0, x)$.

Let $X_l$ denote the output of the $l^{th}$ layer, and $\mathbf{0}$ denote the zero vector. Notice the following properties:
(i) if $X_l = \mathbf{0}$ then the same would be true for all $X_k$ with $k > l$.
(ii) since $P(x_{j,l} = 0) \geqslant P(\forall k : a_{j,k}^{(l)} < 0) = 0.5^w > 0$ we get $P(X_l = \mathbf{0}) > 0$.
Taken together we conclude the following:
as $l \to \infty$ we have $P(X_l = \mathbf{0}) \to 1$, and the output is a simple constant regardless of the input.

The example above presents the core ideas and notation, while keeping the analysis pretty straightforward (thanks to the positive mass on $P(x_{j,l} = 0)$ due to the activation function). We now show that networks with other activation functions (with zero mass on $P(x_{j,l} = 0)$, but a finite derivative at zero) exhibit similar phenomena.

**Definition 1. Asymptotic stability**. Let $X_0, X_1, X_2...$ denote the states of a dynamical system, with $\|X\|$ as the Euclidean norm. We say that a point $X^*$ is *asymptotically stable* if there exists a $\delta > 0$ such that for all $X_0$ in the $\delta$-neighborhood of $X^*$ we have $X_l \to X^*$ as $l \to \infty$.

**Theorem 2 (Asymptotic stability in deep networks)** *In the setting above, consider a deep network with activation function* $\sigma\left(\sum_{k=1}^{w} a_{j,k}^{(l)} x_{k,l-1}\right)$ *with* $\sigma(0) = 0$ *and a finite derivative at zero,* $\sigma_0' < \infty$. *If the weights* $\{a_{j,k}^{(l)}\}_{j,k,l \geqslant 1}$ *are i.i.d. zero-mean with standard deviation smaller than* $\frac{1}{\sqrt{w}\sigma_0'}$ *then* $\mathbf{0}$ *is asymptotically stable.*

*Proof* (sketch). Consider the dynamical system linearized at $\mathbf{0}$. Its Jacobian matrix, $J$, is a $w \times w$ random matrix with i.i.d. zero-mean entries with variance $< \frac{1}{w}$. Now:

- Large width case: for large $w$, according to the circular law (see theorem 1.10 in Tao et al. (2010)), the eigenvalues of $J$ lie in the unit disk with high probability. Standard theory (see, e.g., Boxler (1995) for additional details) entails that $\mathbf{0}$ is a (local) attractor and that for any initial point (close to $\mathbf{0}$) the system will converge to $\mathbf{0}$.

- Fixed width case: although less elegant, it is still possible to bound the Lyapunov exponent of the system, $\lambda_1 \leqslant \frac{1}{2}(ln\frac{1}{w} + ln2 + \psi(w/2))$ where $\psi$ is the digamma function (see details and definitions in chap. 2 sec. 4 in Crisanti et al. (1993), and Cohen & Newman (1984)). Since $\psi(z) \approx lnz - \frac{1}{2z} - O(\frac{1}{z^2})$ we have $\lambda_1 \leqslant -\frac{1}{w} - O(\frac{1}{w^2}) < 0$. Thus, again we conclude from standard theory that $\mathbf{0}$ is a (local) attractor and that for any initial point (close to $\mathbf{0}$) the system will converge to $\mathbf{0}$. ⊠

*Remark 1*: A similar result was obtained by Xiao et al. (2020) for infinitely-wide networks (the crux of the methods in ref. Xiao et al. (2020), which requires the large width, is the use of the central limit theorem to approximate the pre-activations as a gaussian). Our result, in contrast, is not limited to wide networks, and applies to "narrow" networks as well; and although the sketch of the proof mentioned the circular law, and thus a wide network with large $w$ is implied, this is actually not essential. This is merely the most immediate way to bound the eigenvalues of $J$ below 1 in magnitude, and any other way would suffice (see, for example, the second part addressing fixed width). Furthermore, this sheds light on the $O(\frac{1}{\sqrt{w}})$ scaling of $\{a_{j,k}^{(l)}\}_{j,k,l \geqslant 1}$ required.

*Remark 2*: In the next section we also discuss more relaxed and realistic versions of Theorem 2; in particular, including biases and finite-depth networks.

Theorem 2, however, primarily acts as an initial starting point, mainly for didactic purposes. A more interesting question to consider is: Can we establish for deep neural networks a counterpart to Theorem 1 and proposition 2 that sheds light on the behavior after training with examples? Our naive algorithm for training deep neural networks is similar to before: initialize the weights from a normal distribution, and check if the outcome fits the sample data. However, we now also consider a *tolerance* for disagreement smaller than a predefined threshold:

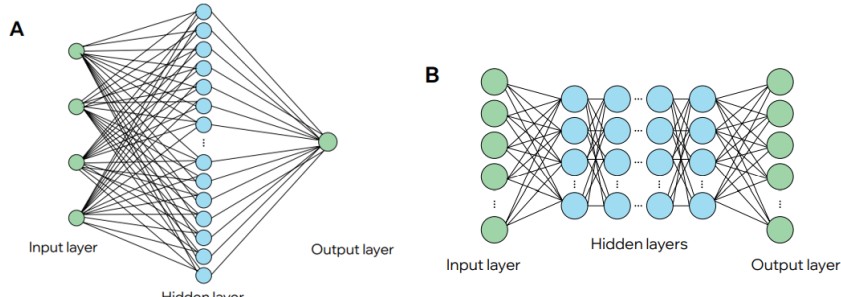

Figure 1: Limiting simplicity for large neural networks. (A) A wide (and shallow) neural network. When the weights are random, summing over the output of the large hidden layer will entail a Gaussian process for the output and thus simplicity. (B) A deep (and narrow) neural network. Here the hidden layers might act as an operator in a dynamical system, driving the initial input towards a fixed point, given as the output of the penultimate layer. Thus, in a deep network the driving force for the emergence of a simple function (the "constant" fixed point) and generalization is different from the one in a wide network.

```
Naive Algorithm (for neural networks).  Given (X,Y_f) and a
tolerance, τ:

    1. Initialize the weights of a network (f̂) at random from a
       normal distribution with variance 1/w.

    2. Check if the network f̂ agrees with f over (X,Y_f) up to τ;
       i.e., each output given by the network on the sample data is
       within a distance τ from the correct output.  If so, return
       f̂.  If not, go to step 1.
```

Note that for typical networks, with a continuous activation-function, the resulting network is continuous - both with respect to its input and moreover with respect to its weights[6]. From this follows:

**Proposition 3** *If $f$ can be described over the sample-data by the neural network (i.e., there is a choice of the weights for the neural network such that it fits the sample-data perfectly), then the naive algorithm terminates successfully in finite time with probability 1.*

*Proof.* Due to continuity with respect to weights, the set of "perfect fit" weights is contained in a ball with radios $\delta_\tau > 0$, i.e. having a strictly positive mass. Thus, the number of attempts by the naive algorithm is dominated by a geometric random variable.

*Remark*: throughout the rest of the paper we will assume that $f$ can indeed be described over the sample-data by the neural network.

Consider the following conditions regarding the network-structure and the data:

- N1: the network is a fully connected convolutional neural network. Let $d$ denote the dimension of the input $x_j$; each layer in the network has also width $d$.
- N2: the activation function is linear near the origin; for example, a shifted hard sigmoid: $\max(-1, \min(1, x))$.
- S1: small sample size. We assume the sample size is smaller than $d$, i.e. $d > |(X, Y_f)|$.
- S2: orthonormal samples. For each $x_i, x_j$ in the sample, we have that $x_i$ and $x_j$ are orthogonal (and have unit length).

Assume the existence of an interpolating solution (i.e., a set of weights with which the network fits the data perfectly) possessing the following properties, and that the naive algorithm succeeds in finding it:

---

[6]For brevity, we focus on regression, though a continuity argument could be made for classification as well.

- A1: Similar length mapping. Assume that for each $x_i, x_j$ in the sample, each one of the layers in the network produces for $x_j$ an output which is roughly the same length, $c_l$, as for $x_i$ (this will be made clear below, and further discussed in the discussion section).
- A2: Quasi-linear prologue. Assume that for each input near the origin, the first layers in the network act on it linearly; only at layer $l_\boxtimes$, with $1 \ll l_\boxtimes$, the non-linearity first takes its effect (this too will be made clear below, and further discussed in the discussion section).

*Remark.* Notice that if a network with depth $m$ is an interpolating solution, then there is an interpolating solution with depth $m + l_\boxtimes$ satisfying assumptions A1-A2; for example, a solution where the first $l_\boxtimes$ layers act as the identity function on the training data.

**Theorem 3 (Probabilistic nearest-neighbor classification)** *Denote the $s$ samples $(X, Y_f)$ by $X = \{x_1, x_2, ...x_s\}$ and $Y_f = \{y_1, y_2, ...y_s\}$. Assume a neural network satisfying conditions N1-S2 is trained with the naive algorithm, and a solution satisfying assumptions A1-A2 is found.*

*Let $x_{new}$ denote an out-of-sample data-point and $y_{new}$ the output by the network corresponding to it.*

*With high probability (tending to 1 as $l_\boxtimes \to \infty$) we have:*
*i) $y_{new} \in \{y_1, y_2, ...y_s\}$.*
*ii) For $y_j \in \{y_1, y_2, ...y_s\}$, $P(y_{new} = y_j)$ is an increasing function of the cosine similarity between $x_{new}$ and $x_j$.*

*Proof.* Let $x_j^{(1)}$ denote the output of the first layer of the network (after training) given $x_j$ as input; more generally, let $x_j^{(k)}$ denote the output of the $k^{th}$ layer of the network given $x_j$ as input.

**Lemma 1** *Let $x_{ort}$ be a new input, orthogonal to the training samples $\{x_1, x_2, ...x_s\}$. Then the output of the first layer, $x_{ort}^{(1)}$ is a zero-mean normally distributed random variable, independent of $\{x_1, x_2, ...x_s\}$ and independent of $\{x_1^{(1)}, x_2^{(1)}, ...x_s^{(1)}\}$*

(see proof in the appendix).

Consider the orthogonal decomposition of $x_{new}$:

$$x_{new} = \sum_{j=1}^{d} h_j x_j = \sum_{j=1}^{s} h_j x_j + h_{ort} x_{ort} \tag{2}$$

Based on A1-2 and lemma (1) we have

$$x_{new}^{(1)} = \sum_{j=1}^{s} h_j c_1 x_j^{(1)} + z \tag{3}$$

where $z$ is a zero-mean normally distributed random variable independent of $\{x_1^{(1)}, x_2^{(1)}, ...x_s^{(1)}\}$.

Clearly, if for some $j$ we have $x_{new}^{(1)}$ equal to $x_j^{(1)}$, or in the $\delta_\tau$ neighborhood of $x_j^{(1)}$, then $y_{new} = y_j$. For which $j$ is this most likely?

**Proposition 4** *The larger the magnitude of $h_j$, the more likely it is that $x_{new}^{(1)}$ is in the $\delta_\tau$ neighborhood of $x_j^{(1)}$.*

Without loss of generality, assume that $c_1 = 1$ (just to simplify the notations). Pick some index, $k$, and rewrite (3) in the following manner:

$$x_{new}^{(1)} = z + \sum_{j=1}^{s} h_j x_j^{(1)} = z + \sum_{\substack{j=1 \\ j \neq k}}^{s} h_j x_j^{(1)} + h_k x_k^{(1)} = z + \left( \sum_{\substack{j=1 \\ j \neq k}}^{s} h_j x_j^{(1)} + (h_k - 1) x_k^{(1)} \right) + x_k^{(1)}$$

In order for $x_{new}^{(1)}$ to be in the $\delta_\tau$ neighborhood of $x_k^{(1)}$ the terms inside the parentheses need to be canceled out by $z$. Since the Euclidean norm of the (vector) term in the parentheses is $\sum_{j=1}^s h_j^2 + 1 - 2h_k$, while $z$ is concentrated at the origin (i.e., a zero-mean unimodal spherically symmetric random variable) we conclude that the larger the magnitude of $h_k$, the more likely it is that $x_{new}^{(1)}$ is in the $\delta_\tau$ neighborhood of $x_k^{(1)}$.

Next, if $x_{new}^{(1)}$ is not within the $\delta_\tau$-neighborhood of any $x_k^{(1)}$, we should examine the probabilities of $x_{new}^{(2)}$ being within the $\delta_\tau$-neighborhood of certain $x_k^{(2)}$. Unfortunately, unlike the case with lemma (1), $x_{ort}^{(2)}$ is not a normally distributed[7]. However, $x_{ort}^{(2)}$ is nevertheless a zero-mean unimodal spherically symmetric random variable; thus, similarly to proposition (4), we conclude that the larger the magnitude of $h_k$, the more likely it is that $x_{new}^{(2)}$ is in the $\delta_\tau$ neighborhood of $x_k^{(2)}$. Furthermore, based on A1-2, the same applies to layers $3, 4, ... l_\boxtimes$, and thus as $l_\boxtimes \to \infty$ we conclude that with high probability:
i) $y_{new} \in \{y_1, y_2, ...y_s\}$.
ii) For $y_j \in \{y_1, y_2, ...y_s\}$, $P(y_{new} = y_j)$ is an increasing function of the cosine similarity between $x_{new}$ and $x_j$.

**Corollary 1** *If a large ensemble of networks is trained under the conditions of theorem 3 then the ensemble-average of their output for $y_{new}$ is a kernel-machine.*

## 3 DISCUSSION

Although Boolean networks and the deep neural networks in Theorems 2-3 are perhaps the simplest "overparameterized" models, and are much simpler than other practical models, we feel there are a few lessons to be learnt from Theorem 1 and Theorems 2-3.

**Shannon effect vs. the 'no Shannon effect'.** Unlike the intuitive *Shannon effect* (i.e., that most Boolean functions are complex (Riordan & Shannon, 1942)), the so-called *no Shannon effect* in *randomly constructed* functions (Genitrini et al., 2014) is virtually unknown in the disciplines of machine learning and statistics. Nevertheless, we hypothesize it plays a key role not only in explaining the generalization of binary neural networks, but also more broadly for other overparameterized models (albeit through analogous phenomena). Indeed, in (Valle-Perez et al., 2018) the authors rightly argued that general Kolmogorov complexity reasoning entails simplicity bias; however, their argument was non-constructive and abstract, whereas below we argue that simplicity bias is not much more than a nearly-inevitable outcome of the central limit theorem (for wide networks) or a dynamical system fixed-point theorem (for deep networks).

**Wide vs. deep networks.** There are two very different mechanisms that bias towards simplicity in neural networks. In a very wide network (see Fig. 1A) with random weights, the situation is governed by the central limit theorem, and thus the network is a Gaussian process (at initialization), which is a simple and well-behaved function of the input.

The main driving force for the emergence of these simple functions in the setting above is the large *width* of the network, and not its depth (Lee et al., 2019). For a "narrow" but *deep* neural network we present an entirely different mechanism that produces simplicity bias nevertheless. In a multi-layered network (see Fig. 1B) each layer can be viewed as an operator in a dynamical system, or Markov chain, that acts on the output of its preceding layer; this should lead to convergence to a fixed point regardless of the initial input - i.e. a simple "constant function".

The aforementioned viewpoint of a Markov-chain "forgetting" its initial condition (i.e., input) elucidate also why adding biases in Theorem 2 would not change the result substantially. Adding external biases, unrelated to the initial input, should not prevent forgetting the initial condition and converging to an output independent of the input (although now instead of having **0** as the output, the output is drawn from the stationary distribution). Similarly, the asymptotic phrasing of Theorem 2 does not imply it is irrelevant for finite-depth networks; on the contrary, like many other "asymptotic conv-

---

[7]Interestingly, when the width of the layers is rapidly decreasing, similarity to autoencoders, subsequent outputs are again near-normal Li & Woodruff (2021).

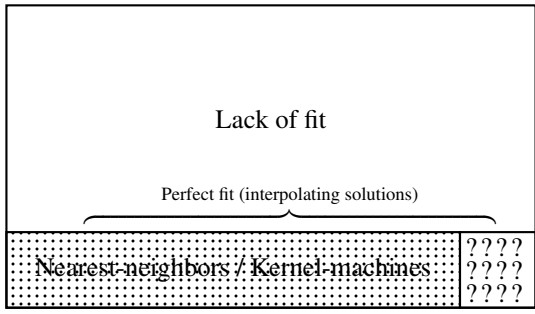

Figure 2: Network space. With random weights, most networks would not fit the sample (white area). Among the networks that do fit the data perfectly, most are simple Nearest-neighbors / Kernel-machines (dotted area). We conjecture that only a small fraction of solutions are more involved.

erence" results, here too convergence to the fixed-point up to an $\epsilon$-tolerance distance from it, occurs within a finite depth (possibly exponentially fast).

Note also that our new conjecture is different and complementary to the *edge of chaos* hypothesis (Langton, 1990), which states that in order for a (post-training) dynamical system to carry out "computations", it needs to be between ordered and chaotic, i.e. at the edge of chaos (e.g., with a Lyapunov exponent $\approx 0$ in absolute value). Our new hypothesis states that a *pre-training* system needs to be in an ordered state (e.g., with a Lyapunov exponent $< 0$ in absolute value) for it to generalize well after seeing the data (see also Xiao et al. (2020); Schoenholz et al. (2016) for additional discussion in a specific setting).

Regardless to the universality and veracity of our conjecture above, the following (known) cautionary tale regarding research on generalization in *wide* networks should be reiterated: generalization in *deep* networks is likely to be driven by a different mechanism, and thus insights from shallow-and-wide networks might not be relevant.

**Training and optimizers.** In addition to providing a concrete and rigorous example of simplicity bias and its contribution to learning, Theorem 1 also suggests a lack of optimizer-dependence. The continuous neural network analogous to the (computationally inefficient) naive algorithm would be "many initializations plus early stopping[8]", suggesting that the role of the specific optimizer is not crucial (and Theorem 3 demonstrates it for regression for an almost-perfect fit). As proposition 2 also implies, the optimizer clearly affects which representation will be sampled (i.e., which function will be obtained after training), but it is not the driving force behind simplicity bias.

Nevertheless, there is certainly merit in research studying specific optimizers. Here we did not address training or the standard optimizers; in Theorem 1 we only addressed learning via "toy training" mostly as a proof of concept - that good learning can be performed in overparameterized models merely via the build-in simplicity bias (and without the use of an optimizer). Additionally, Theorem 2 does not address learning at all, it only serve to show that (randomly initialized) deep networks are biased towards simple functions; nevertheless, there is reason to think that this prior bias will affect the posterior obtained after training. This seem even more likely when considering training which start at a random initialization and is updated via iterations of a local search (as most common optimizers do). Indeed, ref. Xiao et al. (2020) demonstrated similar results for infinitely-wide deep networks (while here we also address finite-width, or "narrow", networks).

Theorem 3 show that random constructions that fit the sample act as simple nearest-neighbors classification or kernel-machines, suggesting that perhaps the vast majority of interpolating solutions behave so as well (see fig. 2). After a random initialisation, SGD merely changes the weights greedily to find the closest solution, without any ingenious hidden addition; thus, we conjecture that the solutions SGD finds are of similar nature (see Domingos (2020)), while "sophisticated" solutions are very rare.

---

[8]Note that for classification, unlike regression, even if the weights are continuous random variables, there is a non-zero probability of random weights yielding a network that fits the training data perfectly (albeit possibly a very small probability, and subject to the existence of such a fit).

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

## A    APPENDIX

Here we provide the proof of lemma 1.

Recall the following affine transformation theorem:

**Theorem 4 (Affine Transformation Theorem)** *Let $X$ be a $p$-dimensional multivariate normal random variable, i.e., $X \sim \mathcal{N}(\mu, \Sigma)$, where $\mu$ is the mean vector and $\Sigma$ is the covariance matrix of $X$. Let $A$ be a constant $q \times p$ matrix. Then, the random variable $Y = AX$ is also multivariate normally distributed, and its distribution is given by:*

$$Y \sim \mathcal{N}(A\mu, A\Sigma A^T)$$

Rearrange the (transpose) of the inputs, $x_j^T$, and $x_{ort}^T$ in the following block matrix:

$$
\begin{bmatrix}
x_1^T & 0 & \cdots & 0 \\
0 & x_1^T & \cdots & 0 \\
\vdots & \vdots & \ddots & \vdots \\
0 & 0 & \cdots & x_1^T \\
x_2^T & 0 & \cdots & 0 \\
0 & x_2^T & \cdots & 0 \\
\vdots & \vdots & \ddots & \vdots \\
0 & 0 & \cdots & x_2^T \\
\vdots & \vdots & \ddots & \vdots \\
x_s^T & 0 & \cdots & 0 \\
0 & x_s^T & \cdots & 0 \\
\vdots & \vdots & \ddots & \vdots \\
0 & 0 & \cdots & x_s^T \\
x_{ort}^T & 0 & \cdots & 0 \\
0 & x_{ort}^T & \cdots & 0 \\
\vdots & \vdots & \ddots & \vdots \\
0 & 0 & \cdots & x_{ort}^T
\end{bmatrix}
$$

where each "0" is a vector of an appropriate dimension.

Recall that for the first layers, under A2 and our previous notation, the output of $x_{j,l}$, the $j^{th}$ unit in the $l^{th}$ layer, is $\sum_{k=1}^{w} a_{j,k}^{(l)} x_{k,l-1}$. Focusing on the first layer, we drop the superscript notation and write $a_j = (a_{j,1}, a_{j,2}, a_{j,3}, ... a_{j,w})^T$.

Concatenate the weights in the following column vector:

$$
\begin{bmatrix}
a_1 \\
a_2 \\
a_3 \\
\vdots \\
a_w
\end{bmatrix}
$$

obtaining

$$
\begin{bmatrix}
x_1^T & 0 & \cdots & 0 \\
0 & x_1^T & \cdots & 0 \\
\vdots & \vdots & \ddots & \vdots \\
0 & 0 & \cdots & x_1^T \\
x_2^T & 0 & \cdots & 0 \\
0 & x_2^T & \cdots & 0 \\
\vdots & \vdots & \ddots & \vdots \\
0 & 0 & \cdots & x_2^T \\
\vdots & \vdots & \ddots & \vdots \\
x_s^T & 0 & \cdots & 0 \\
0 & x_s^T & \cdots & 0 \\
\vdots & \vdots & \ddots & \vdots \\
0 & 0 & \cdots & x_s^T \\
x_{ort}^T & 0 & \cdots & 0 \\
0 & x_{ort}^T & \cdots & 0 \\
\vdots & \vdots & \ddots & \vdots \\
0 & 0 & \cdots & x_{ort}^T
\end{bmatrix}
\cdot
\begin{bmatrix}
a_1 \\
a_2 \\
a_3 \\
\vdots \\
a_w
\end{bmatrix}
=
\begin{bmatrix}
x_1^{(1)} \\
x_2^{(1)} \\
x_3^{(1)} \\
\vdots \\
x_{ort}^{(1)}
\end{bmatrix}
$$

Recall that $(a_1, a_2, a_3, ...a_w)^T$ is normally distributed with a diagonal covariance matrix, and that it is here multiplied by a matrix with orthogonal rows; thus, by Theorem (4) their product is normally distributed with a diagonal covariance matrix. In particular, $x_{ort}^{(1)}$ is a zero-mean normally distributed random variable, and conditioning on $\{x_1^{(1)}, x_2^{(1)}, ...x_s^{(1)}\}$ does not change its distribution.

