# OpenReview forum: "Simplicity Bias in Overparameterized Machine Learning"
_ICLR.cc/2024/Conference — Submitted to ICLR 2024_

### Official Review · Reviewer_Y87b · 2023-10-26

**Soundness:** 3 good
**Presentation:** 3 good
**Contribution:** 3 good
**Rating:** 3
**Confidence:** 5

**Summary:**

This work aims to understand the simplicity bias for overparameterized learning models. Contrary to the traditional belief that gradient based optimizers play an important role for generalization, the authors claim that simplicity bias and excellent generalization in these models are optimizer independent. Infact, at initialization of these deep models, the functions obtained are uniformly at random constructed and not uniformly at random sampled because of the construction process (architectural constraints). Hence the probability space over the functions is not uniform and infact biased towards certain "simple functions". Using the help of three cases, the authors show their point :
1) Random Boolean tree at initialziation performs better than chance agreement.
2) Wide neural networks at initialization are biased towards Gaussian process conditoned on interpolating the sampled data
3) Deep neural networks with larger depth will have the tendency to have 0 (a constant function) as the asymptotic stable point.

**Strengths:**

1) Paper is well written, easy to understand.
2) The claims made are original (not necessarily true) and the problem is signficantly important.

**Weaknesses:**

In my opinion, the authors have vastly overstated the claim that obtaining good/generalizable *functions* is generic and massively understated the role of gradient based optimizers. This is accompanied by incomplete literature review (that supports their hypothesis for example [1]) and the citations therein. Here are some detailed points:

1) I agree with the authors that randomly initialized neural networks acting as simplicity bias can serve as good generalizable priors which in turn will aid getting good posteriors after training, but the **extent to which this simplicity bias prior plays a bigger role than gradient descent is not clear**. Infact on the contrary, even if *the deep linear overparameterized models are ill-initialized, gradient descent finds a way to find generalizable solutions [2,3]*.

2) Recently, it is quite well understood how finite step-size gradient descent has a self-stabilization mechanism to restrict sharpness of iterates to 2/l.r [4] which is pivotal in biasing iterates to flat landscapes. Infact this property of GD is irrespective of any architecture assumption. Hence comments such as "excellent generalization are optimizer independent" goes against the prior 7-8 years of prior theoertical study on gradient descent.

3) The authors did not support their claims with experiments such as in simple models. How does simplicity bias from the deep model initialization compare to GD's generalization capability. How does one work without the other? How worse can generalization degrade if we do not use gradient based methods (see experiments in [1]). I highly suspect there are cases, when gradient descent based methods will stop generalizing with worse initializations.

4) I still find the observations regarding the simplicity bias to be interesting, however, I do not quite understand how they aid generalization or prevent overfitting. For example, in wide networks, the network is biased to a Gaussian process, but how does that aid generalization ? Similarly, for deep networks, it is easy to see as depth tend to inifinity, 0 becomes the fixed point, but it is merely a bias (not necessarily a good bias).

5) Can the authors comment on how their work is different than https://openreview.net/forum?id=rye4g3AqFm ? They seem to have the same claims and they also show strong simplicity bias in a model DNN for Boolean functions.






[1] Chiang, Ping-yeh, et al. "Loss landscapes are all you need: Neural network generalization can be explained without the implicit bias of gradient descent." The Eleventh International Conference on Learning Representations. 2022.

[2] Wang, Yuqing, et al. "Large learning rate tames homogeneity: Convergence and balancing effect." arXiv preprint arXiv:2110.03677 (2021).

[3] Gradient descent for matrix factorization: Understanding large initialization (https://openreview.net/forum?id=fAGEAEQvRr)

[4] Damian, Alex, Eshaan Nichani, and Jason D. Lee. "Self-stabilization: The implicit bias of gradient descent at the edge of stability." arXiv preprint arXiv:2209.15594 (2022).

**Questions:**

a) I would like the authos to answer point-4 first. As this question is critical to better understand the contribution.
b) I still find the claims of the paper to be overstated and not true based on my point on weakness 1 and 2.
c) Furthermore, the lack of experiments does not help to support the claims.

---

### Official Review · Reviewer_CuZp · 2023-10-31

**Soundness:** 3 good
**Presentation:** 3 good
**Contribution:** 2 fair
**Rating:** 3
**Confidence:** 4

**Summary:**

The paper suggest the so called "simplicity bias" is not purely because of optimization. They suggest that "the probability space over models is in fact biased towards simple functions". They explore the idea for different learning methods such as decision trees, (wide+shallow) and (narrow+deep) neural networks and nearest neighborhood algorithms. They proposed a naive random algorithm that relies solely on the initialization/architecture of the neural network, making it more straightforward to analyze. They demonstrated that this naive random algorithm is not completely random and can avoid overfitting, indicating a potential simplicity bias stemming from the model architecture and its initialization independent of optimizer.

**Strengths:**

1. the paper is well written and easy to understand.
2. They provided some insight that how initialization and architecture can play a role in generalization which I found interesting.

**Weaknesses:**

1. The message of the paper is not clear. The abstract is written in a way that avoids the authors' idea of simplicity bias. It mostly explains what it is not. it is claimed that the authors studied the "source" of simplicity bias but I do not see it in the paper. My question is what the "source" is? It seems based on what is presented the source is initialization and architecture. Can you elaborate this, is my take correct?

2. Second, related to the first point, it is mentioned that "simplicity bias and excellent generalization are optimizer-independent". I believe this is too strong. I do not see any evidence in the paper to support this claim. I understand the point that initialization and architecture can also affect the simplicity bias but I do not agree that it is optimizer-independent. For example consider a deep linear network and simple linear regression in the over-param regime. There exist more than one solution in both cases but because the architecture is different, (S)GD potentially finds different solution in these two settings.

**Most importantly:**

3. Related to 2, your analysis appears to emphasize the importance of the loss landscape and initialization for the optimizer. In fact, one could argue that your naive algorithm essentially functions as an optimizer. Typically, with an optimizer, you have a loss function and aim to minimize it using a specific optimization algorithm. Your naive algorithm seems to generate new random weights in each iteration and evaluates the loss function. If the results aren't satisfactory, the process is repeated until convergence. How does this differ from a typical optimizer mechanism? Of course architecture and initialization play a crucial role in the generalization performance of the solution determined by the optimizer.

**Questions:**

Please see weaknesses.

---

### Official Review · Reviewer_prbm · 2023-11-01

**Soundness:** 2 fair
**Presentation:** 1 poor
**Contribution:** 2 fair
**Rating:** 3
**Confidence:** 3

**Summary:**

The study addresses the insufficient theoretical understanding of why overparameterized models like deep networks have exceptional generalization capabilities. It identifies "simplicity bias" as a significant factor in this context. The research outlines four key points: (i) simplicity bias can justify the generalization seen in models like neural networks; (ii) the presence of simplicity bias and good generalization is not reliant on the type of optimizer used in training; (iii) the way simplicity bias works in pre-trained models is linked to how randomly constructed priors are not truly random in sampling; and (iv) the way bias operates varies between wide, shallow networks and deep, narrow networks.

**Strengths:**

The paper attempts to shed light on the understudied area of simplicity bias in overparameterized models, which could contribute to the ongoing discourse on deep network generalization.

**Weaknesses:**

1. The system models are not clearly described. For example, in deep neural networks depicted in Fig. 1B, the choice of the loss function is not introduced.

2. Some results are stated vaguely. For example, in Claim 1, the meaning of "the vast majority of functions" is unclear.

3. The "Naive Algorithm (for neural networks)" in Page 6 is very confusing. The authors should provide necessary evidence or references that those Naive Algorithms are meaningful in reality.

4. The writing quality should be improved. The authors provide some claims, theorems, and propositions. However, their connections seem very loose.

**Questions:**

1. The last equation in the proof of Theorem 1 seems to have an extra "=".

2. What is point of revisiting the known results for infinitely-wide neural networks?

---

### Official Review · Reviewer_HAQJ · 2023-11-01

**Soundness:** 1 poor
**Presentation:** 1 poor
**Contribution:** 1 poor
**Rating:** 3
**Confidence:** 2

**Summary:**

The reviewed work studies the effect of ``simplicity bias`` the generalization in machine learning algorithms ranging from the tree-based learning of Boolean functions to deep and shallow neural networks.

The argument of the authors is that even though these models are constructed using uniform randomness, the resulting distribution (over all input-output functions) is not random in the usual sense. For a data generating process that is simple (in a sense that plays well with the distribution of the random classifier), this allows for generalization.

The paper proves such a result (relying heavily on prior work) in theorem 1, in the setting of Boolean functions and trees. It further shows that when training an infinitely wide network by conditioning the prior distribution on the training data it obtains the same distribution as when training the network via gradient descent. Finally, the authors show some results on the case of deep but not wide neural networks.

**Strengths:**

The study of the generalization properties of deep (or otherwise) learning is an important objective and the authors involve techniques and ideas from quite distant communities (such as Boolean Trees) that are not commonly used to study this topic.

**Weaknesses:**

I found the paper very difficult to read and am still confused what it's main punchline is supposed to be.

1. Aren't the "naive algorithms" presented throughout the papers just complicated ways of saying: "Use an arbitrary sample from the prior (initialization) distribution, conditional on matching the data?

2. The authors write

> The crux of the matter is that the functions obtained (at initialization) are uniformly-at-random constructed but not uniformly-at-random
sampled. In other words, the naive “default” probability space over functions, where each unique
function is sampled with equal probability, is actually conceptually wrong and irrelevant for this
domain. Instead, it is well-known that if a construction process is used to “sample” complex objects
then the resulting probability space is not a uniform space (even if the construction itself is uniformly
random, i.e., in each sub-step we choose uniformly from several construction options). Moreover,
different (spaces of) construction processes yield different probability spaces over functions. However, several properties can be expected to be universal in some sense, with simplicity bias being a
case in point (Chauvin et al., 2004).

As the authors themselves write, it is well known that the distributions of classifiers that are constructed using uniform randomness are not necessarily uniformly random (in any given sense). "In other words, the naive “default” probability space over functions, where each unique
function is sampled with equal probability, is actually conceptually wrong and irrelevant for this
domain." is thus a strawman.

3. Given that it is well-accepted that generalization has to arise from choosing a non-unformly random classifier in a way that is well aligned with a notion of simplicity/structure in the data, the crux of the matter becomes making this argument more precise and relatating the distributions of classifiers produced by ML algorithms to simplicity found in the data. I don't see how this paper makes any progress on that.

4. For instance, it doesn't strike me as "surprising" that random Boolean functions constructed from Trees would do better at fitting functions that abide by a tree-based simplicity requirement than uniformly Boolean functions.

5. It is my understanding that Proposition 2 just amounts to saying that gradient descent training on infinitely wide limits amounts to taking a conditional distribution according to the Gaussian process arising from the random initialization. This seems plausible and may be the most relevant result, but as far as I can see there is no proof provided?

6. I am honestly at a loss as to the meaning and significance of the results on deep networks. Why are they related to the "no-shannon effect"?  What is the no-shannon effect even?

7. Finally, the empirical success of neural networks seems to rely heavily on training, why are the authors so confident that conditioning the initial distribution can explain generalization of deep learning?

**Questions:**

Please respond to by concerns in the last section.

---

### Meta-Review · Area_Chair_FCGs · 2023-12-05

**Metareview:**

This paper studies the simplicity bias in overparametrized machine learning. The paper tries to demonstrate that training overparametrized models (from tree-based models to shallow neural networks) will generate simple functions due to the structured randomness in the function space. While the reviewers are generally interested in the direction of the result, there is a lot of confusion around what the paper has actually proved, and some have concerns about overclaiming results. The authors should take the reviews into consideration and significantly improve the write-up for future submissions.

**Justification For Why Not Higher Score:**

All reviewers found the paper very confusing.

**Justification For Why Not Lower Score:**

N/A

---

### Decision · Program_Chairs · 2024-01-16

Reject